# Tree Species Classifications of Urban Forests Using UAV-LiDAR Intensity Frequency Data

**Yulin Gong** [1,2,3], **Xuejian Li** [1,2,3], **Huaqiang Du** [1,2,3,*] , **Guomo Zhou** [1,2,3], **Fangjie Mao** [1,2,3], **Lv Zhou** [1,2,4], **Bo Zhang** [1,2,3], **Jie Xuan** [1,2,3] **and Dien Zhu** [1,2,4]

1    State Key Laboratory of Subtropical Silviculture, Zhejiang A & F University, Hangzhou 311300, China
2    Key Laboratory of Carbon Cycling in Forest Ecosystems and Carbon Sequestration of Zhejiang Province, Zhejiang A & F University, Hangzhou 311300, China
3    School of Environmental and Resources Science, Zhejiang A & F University, Hangzhou 311300, China
4    The College of Forestry, Beijing Forestry University, Beijing 100083, China
*    Correspondence: duhuaqiang@zafu.edu.cn

**Abstract:** The accurate classification of tree species is essential for the sustainable management of forest resources and the effective monitoring of biodiversity. However, a literature review shows that most of the previous unmanned aerial vehicle (UAV) light detection and ranging (LiDAR)-based studies on fine tree species classification have used only limited intensity features, accurately identifying relatively few tree species. To address this gap, this study proposes developing a new intensity feature—intensity frequency—for the LiDAR-based fine classification of eight tree species. Intensity frequency is defined as the number of times a certain intensity value appears in the individual tree crown (ITC) point cloud. In this study, we use UAV laser scanning to obtain LiDAR data from urban forests. Intensity frequency features are constructed based on the extracted intensity information, and a random forest (RF) model is used to classify eight subtropical forest tree species in southeast China. Based on four-point cloud density sampling schemes of 100%, 80%, 50% and 30%, densities of 230 points/m$^2$, 184 points/m$^2$, 115 points/m$^2$ and 69 points/m$^2$ are obtained. These are used to analyze the effect of intensity frequency on tree species classification accuracy under four different point cloud densities. The results are shown as follows. (1) Intensity frequencies of trees are not significantly different for intraspecies ($p > 0.05$) values and are significantly different for interspecies ($p < 0.01$) values. (2) The intensity frequency features of LiDAR can be used to classify different tree species with an overall accuracy (OA) of 86.7%. *Acer Buergerianum* achieves a user accuracy (UA) of over 95% and a producer accuracy (PA) of over 90% for four density conditions. (3) The OA varies slightly under different point cloud densities, but the sum of correct classification trees (SCI) and PA decreases rapidly as the point cloud density decreases, while UA is less affected by density with some stability. (4) The priori feature selected by mean rank (MR) covers the top 10 posterior features selected by RF. These results show that the new intensity frequency feature proposed in this study can be used as a comprehensive and effective intensity feature for the fine classification of tree species.

**Keywords:** tree species classifications; unmanned aerial vehicle (UAV); LiDAR; point cloud; intensity frequency

## 1. Introduction

The classification of tree species for research on forest inventory [1], carbon storage assessment [2], habitat [3], and ecosystem changes [4] is crucial. Remote sensing data have characteristics such as simultaneous monitoring and repeatability over large areas, which can provide landscape-scale perspectives and forest structure information [5], offering an excellent advantage for large-scale tree monitoring, mapping, and assessment. Therefore, remote sensing technology has become essential for interpreting biodiversity information. How to classify tree species from remote sensing data has been a trending research topic both in China and abroad.

Obtaining forest tree species information through remote sensing usually requires direct and indirect methods [6]. The direct methods directly classify species, community types, and abundance with data based on high spatial or spectral resolution. The spatial resolution of the data often reaches the individual tree scale or even the leaf scale to perform ITC detection [7–9]. Studies have shown that tree species classification accuracy of more than 80% can be achieved by using data with spatial resolution at the leaf scale and spectral resolution of 3–10 nm [8,10]. The indirect methods refer to the acquisition of derived variables such as the normalized difference vegetation index (NDVI) and the enhanced vegetation index (EVI) from remote sensing data [11]. The variables are then combined with field data to construct models to classify tree species. Light detection and ranging (LiDAR) is a remote sensing technology that can be employed to obtain the three-dimensional geographic coordinates of a research object, especially information relating to vertical structure, directly, quickly and accurately [12]. It can also guarantee the efficient and accurate monitoring of a target from different spatial scales [13]. LiDAR works by recording the transmitted and echoed signals of high-frequency pulses capable of penetrating the vegetation canopy gaps [13]; these signals are then compared and appropriately processed to obtain parameters such as the distance, coordinates, and height of the target [14]. Therefore, LiDAR can be used for studies on tree height [15], canopy density [16], leaf area index [17], individual tree detection [18,19] and tree species classification [20].

From a sensor perspective, there are three common approaches to tree species classification when using LiDAR: using a single LiDAR to extract features such as structure and intensity to classify tree species; using multiple LiDAR to extract structural and intensity features, vegetation index features, etc., to classify tree species; and using LiDAR with other sensors (e.g., multispectral, hyperspectral) to extract structural features, spectral features, vegetation index features, etc., to classify tree species. The first data processing approach is relatively simple, and the potential errors in data caused by sensor differences can be avoided, but the classification accuracy is limited. For example, Ørka et al. (2007) [21] constructed mean intensity features and intensity standard deviation features of ITC based on multi-echo data from a single LiDAR and classified tree species by principal component analysis and linear discriminant analysis with a classification accuracy of 68–74%. Vaughn et al. (2012) [22] used a support vector machine (SVM) to classify five tree species in the Pacific Northwest of the United States based on the structural information and average intensity information provided by discrete point LiDAR data, with an overall accuracy of 79.2% (kappa = 0.74). Hamraza et al. (2019) [23] also used LiDAR intensity information to classify needles and broadleaf trees, and their classification accuracy was 65–90%. The second approach, i.e., based on a combination of multiple LiDAR data, has greater potential for accuracy and quantity of tree species classification but also suffers from relatively cumbersome data collection and processing, in addition to potential errors due to discrepancies in data from different sources. For example, Korpela et al. (2010) [24] realized the classification of three tree species in Finland based on two kinds of LiDAR data, with an accuracy of 88–90%. Vaughn et al. (2012) [22] combined airborne waveform LIDAR with discrete point LIDAR to improve the overall accuracy from 79.2% (kappa = 0.74) to 85.4% (kappa = 0.817) for five species. Brindusaet (2018) [25] constructed intensity features and vegetation index features to classify ten tree species based on intensity information from LiDAR at 1550 nm, LiDAR at 1064 nm, and LiDAR at 532 nm with an accuracy of 57.6–90.6%. The third approach uses LiDAR with other sensors (e.g., multispectral, hyperspectral) to classify tree species [26]. This method can fully obtain vegetation spectral and structural information to classify tree species accurately. However, LiDAR data are often used as secondary information. Combining the homo-spectral and hetero-spectral features of optical remote sensing data limits the accuracy of tree species classification to some extent. For example, Shi et al. (2018) [27] used a random forest classification model to classify 15 common urban tree species by combining structural parameters extracted from LiDAR data and a vegetation index extracted from hyperspectral data, with an overall accuracy of 70%. Kukkonen et al. (2019) used multispectral and airborne LiDAR

data to predict three tree species in eastern Finland, with an overall accuracy of 88.2% (kappa = 0.79) [28]. Moreover, hyperspectral sensors are expensive compared with LiDAR sensors, and data processing is more complicated.

In addition to precise 3D coordinates, most LiDAR systems also record "intensity", loosely defined as the strength of the backscattered echo for each measured point. LIDAR intensity data have proven beneficial in species community classification and richness estimation because they are related to surface parameters such as reflectance [29,30]. However, a literature review shows that most previous studies on the fine classification of tree species based on UAV LiDAR based on limited intensity features and fewer tree species. Korpela et al. (2010) [24] and Heinzel and Koch (2011) [31] derived intensity features such as Imean, Istd, Iskewness and Ikurtosis, which represent the mean, standard deviation, skewness and kurtosis of the intensity values for all of the laser points within a crown. Tomohiro et al. (2017) [32] derived intensity image features, which represent RGB images generated using LiDAR intensity. Brindusaet et al. (2018) [25] derived PE, IR, D1_2, RM, and NDIR, which represent the intensity value at percentiles, the standard deviation of the mean intensity of each ring after dividing the tree point cloud into rings, the difference of intensity between the first and second returns of the same pulse in the same channel, the ratio between different statistics, and the infrared normalized difference vegetation index. Therefore, how to use LiDAR intensity for fine tree species classification is a problem worth studying. To further explore the intensity of data-based tree species classification, we propose the concept of intensity frequency. Intensity frequency is defined as the number of times a certain intensity value appears in the ITC point cloud.

Currently, the single-echo scanning speed for mainstream commercial LiDAR ranges between 100,000 and 500,000 points/s. Reducing the number of laser points emitted per second can reduce the power consumption per unit time and improve the efficiency of data acquisition, but it will affect the intensity frequency of ITC.

In this study, eight major tree species in the study area are used as objects, and 147 intensity frequency features are constructed based on unmanned aerial vehicle laser scanning (UAV) data for tree species classification using a random forest (RF) model. The accuracy of tree species classification at four different point cloud densities is analyzed.

Our overall aim was to gain an understanding of the species discrimination potential of intensity frequency features. The specific research aims were to:

1. Describe the differences of intensity frequency between interspecies and intraspecies;
2. Demonstrate the ability of intensity frequency feature in tree species classification;
3. Quantify the effects of point cloud density in species classification;
4. Examine the consistency between feature selected by MR and feature selected by RF.

## 2. Study Area and Method

### 2.1. Study Area

The study area is located on the East Lake campus of Zhejiang A&F University, Lin'an District, Hangzhou City, Zhejiang Province, China, as shown in Figure 1. The East Lake campus covers an area of 1.3 km$^2$. The main tree species include *Ginkgo Biloba*, *Acer Buergerianum Miq*, *Cinnamomum Camphora*, *Magnolia Grandiflora*, *Celtis Sinensis Pers*, *Glyptostrobuspensilis*, *Michelia Figo*, and *Salix Babylonica*, which are common green tree species in cities in subtropical regions.

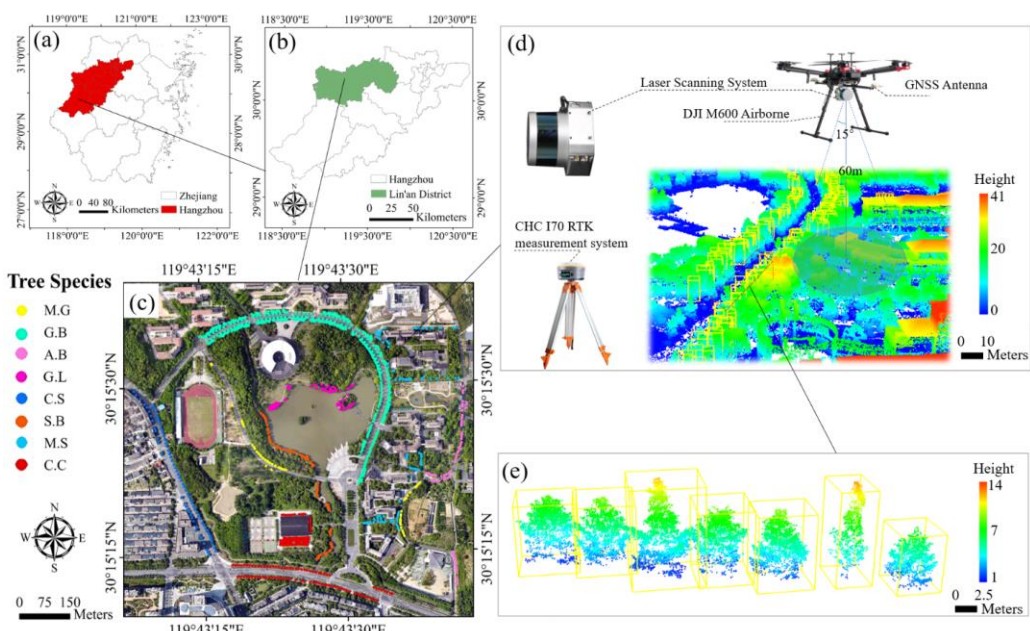

**Figure 1.** (**a**,**b**) show the geographical location of the study area. (**c**) is the distribution of the eight tree species. (**d**) is the diagram of LiDAR data acquisition, and (**e**) is a schematic diagram of the LiDAR slices of G.B.

## 2.2. Field Data

An urban forest is mainly characterized by blocks, strips, and individual trees, with a fragmented distribution and heterogeneous understory, which is very different from a large and continuously distributed forest in the general sense [33,34]. In this study area, there are more trees in a strip distribution, and the understory vegetation is mostly low shrubs and herbaceous plants. Based on the above characteristics, 19 sample strips were established in the study area for ground surveys using a UAV. The ground survey classified 1106 trees and eight species, namely, *Ginkgo Biloba* (G.B: n = 489), *Michelia Figo* (M.F: n = 91), *Cinnamomum Camphora* (C.C: n = 191), *Celtis Sinensis Pers* (C.S: n = 56), *Acer Buergerianum Miq* (A.B: n = 73), *Salix Babylonica* (S.B: n = 91), *Glyptostrobuspensilis* (G.L: n = 68), and *Magnolia grandiflora* (M.G: n = 47). Standing trees with a diameter at breast height (DBH) greater than 5 cm were measured in each strip. Measurements included individual tree position, species, height, canopy height, and crown radius due north and south. The DBH of the individual trees was measured using a DBH ruler. We use two real time kinematics (RTKs) of the same model (CHC i70), one as a reference station and one as a mobile station to measure the position of the tree. Treetop height was measured using a Blume–Leiss altimeter. Since UAV observations are carried out from top to bottom, there is a small probability that individual trees in the middle and low layers will be detected by a single-echo UAV [35,36]. The lower and middle vegetation is often not the same species as those studied here, which will directly increase the intensity complexity and make tree species classification more difficult. Therefore, these individual trees were automatically excluded from the data analysis and tree species classification process of this study.

## 2.3. LiDAR Data Acquisition and Processing

In this study, a DJI Matrice 600 Pro six-rotor UAV was used as the remote sensing platform, equipped with a lightweight Velodyne Puck LITE™ LiDAR sensor and combined with a ground CHC i70 RTK receiver measurement system for data acquisition, as shown in Figure 1d. Due to equipment constraints, the main LiDAR data were obtained on 17 April 2021, and supplements were obtained on 15 May 2021. The tree species G.B, M.F, C.C, A.B, S.B, G.L, and M.G are segmented as ITC using LiDAR data from 17 April; C.S is segmented as ITC using LiDAR data from 15 May. During the flight, the UAV had an

average flight height of 60 m, an average flight speed of 8 m/s, and a flight path spacing of approximately 25 m, and the lateral overlap rate of data sampling was approximately 50%. The sensor recorded a laser pulse wavelength of 903 nm, maximum scanning angle of $\pm 15°$, scanning frequency of 20 Hz, and scanning speed of 300,000 points/s. Considering that the multiple echoes of LIDAR may cause interference in the intensity frequency, we only selected the first echo information. The average point cloud density was obtained using the total number of point clouds after denoising, divided by the area, which was 230 points/m$^2$. The detailed LiDAR parameters are listed in Table 1.

**Table 1.** Summary of remote sensing data acquisition information.

| Parameters | Information | Parameters | Information |
| --- | --- | --- | --- |
| Sensor | Velodyne Puck LITE ™ | Ranging Accuracy | 3 cm |
| Date of Acquisition | 2021.4.17, 2021.5.15 | Mean Point Density | 230 points/m$^2$ |
| Height | 60 m | Wavelength | 903 nm |

As shown in Figure 2, the preprocessing of LiDAR data includes the original point cloud decomposition, denoising, separation of ground points, and normalization. The process is as follows: The LiDAR data, GNSS antenna data, and ground CHC i70 RTK data are processed in the accompanying commercial software Inertial Explorer, and then the LAS point cloud is exported in ZtLiDAR V2.2.0 software. In this paper, the point cloud format is LAS1.4. Then, the noises were denoised, including the very high point caused by other low-flying UAVs or birds and the very low point formed by the computational anomalies, and then the denoised point clouds were separated from the ground points and normalized in the LiDAR360 software.

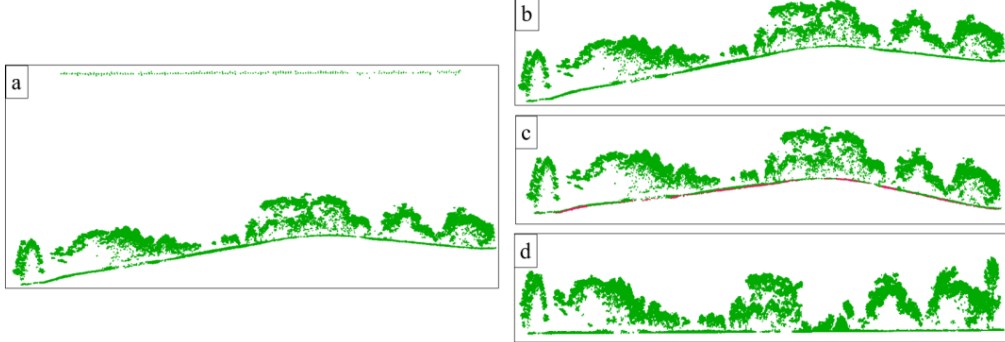

**Figure 2.** (**a**) Slice of the point cloud after decoding the raw data; (**b**) slice of the point cloud after denoising; (**c**) slice of the separated ground point cloud; and (**d**) slice of the normalized point cloud.

### 2.4. ITC Information Acquisition

#### 2.4.1. Extraction of ITC

In this study, the point cloud segmentation (PCS) algorithm was used for ITC extraction [37]. It is a top-to-bottom region-growing approach used to segment trees individually and sequentially from the point cloud [8]. The algorithm starts from a tree top and "grows" an individual tree by including nearby points based on the relative spacing. Points with a spacing smaller than a specified threshold were classified as the target tree, and the threshold was approximately equal to the crown radius [8]. Additionally, the shape index (SI) was added to improve segmentation accuracy by avoiding the elongated branch. The PCS algorithm was implemented using LiDAR360 software, and the space threshold was set as the corresponding average crown diameter of each tree species. The segmentation result was a single point cloud cluster tree, and each point cloud in the same cluster tree had the same tree ID. Finally, the crown of each tree was obtained by filtering ground points and low vegetation points through height filtering, and the height filtering threshold was set to 2 m.

The accuracy of segmentation is evaluated by using the following three metrics: recall (*r* represents the tree detection rate in Equation (1)), precision (p represents the precision of detected trees in Equation (2)), and F1-score (F1 presents the overall accuracy taking both omission and commission into consideration in Equation (3)).

$$r = \frac{\text{Nt}}{\text{Nt} + \text{No}} \tag{1}$$

$$p = \frac{\text{Nt}}{\text{Nt} + \text{Nc}} \tag{2}$$

$$F_1 = 2 \times \frac{\text{r} \times \text{p}}{\text{r} + \text{p}} \tag{3}$$

where Nt is the number of objects correctly identified as trees, No is the number of trees omitted by the PCS algorithm, and Nc is the number of objects wrongly identified as trees.

2.4.2. Intensity Correction of ITCs and Resampling

The LiDAR intensity depends on the power of the backscattered laser pulse measured by the sensor [38]. According to the radar equation, during transmission, the LiDAR intensity is mainly affected by the spectral reflectance of the surface ($\rho$), the sensor receiving the emitted laser power ($P_E$), the LiDAR optical transmission characteristics ($\eta_{sys}$), the distance (R), the angle of incidence ($\theta$), the receiver aperture diameter (D), and the atmospheric attenuation ($\eta_{atm}$) [39]. To reduce the intensity errors in the transmission process caused by the above factors, we model the relationship between the intensity values and the system variables from the LiDAR equation. For the extended Lambert scatterer, the LiDAR equation can be simplified to Equation (4) [40].

$$P_r = \frac{\pi P_E \rho D^2}{(4R^2)} \eta_{atm} \eta_{sys} |\cos \theta| \tag{4}$$

where $\rho$ is the spectral reflectance of the surface. $P_E$, D, $\eta_{sys}$, and $\eta_{atm}$ are assumed to be constant C during the same flight [41]. This leads to Equation (5).

$$P_r = C \frac{\rho}{R^2} |\cos \theta| \tag{5}$$

$$C = \frac{\pi P_E D^2}{16} \eta_{atm} \eta_{sys} \tag{6}$$

From Equation (5), the LiDAR equation relates the received power $P_r$ to the system variables (R, $\theta$), so the relationship between the intensity value *I* and the system variables can be converted into the relationship between the intensity value and the received power $P_r$. Inside the LiDAR receiver, $P_r$ is finally converted into a calibrated integer (digital number, DN), and this integer is the intensity value *I* in raw point cloud data. The logarithmic correction model of the laser intensity value proposed by Tan et al. (2014) [40]. To modify the influence of different incident angles at different distances on the LiDAR intensity the logarithmic correction model is selected. The logarithmic model assumes that the receiver converts the received power logarithmically and then converts it to the laser intensity value, as in Equation (7):

$$I + \tau = K_1 ln p_r + K_2 \tag{7}$$

where *I* is the uncorrected intensity value and $K_1$ and $K_2$ are model coefficients.

The purpose of the intensity correction is to correct the intensity for the variables (R, $\theta$), so that this value depends only on the target property information ($\rho$); therefore, from Equation (7), $P_r$ can be replaced by $\cos \theta / R^2$, and Equation (7) can be written as Equation (8)

$$I + \tau = K_1 \ln \frac{|\cos \theta|}{R^2} + K_3 \tag{8}$$

where $\tau$ is the observation error; $K_3 = K_2 + \ln^C$.

After the logarithmic change of Equation (8), the logarithmic correction model of the LiDAR intensity value can be obtained as Equation (9):

$$I_s = (I - K_3)\frac{\ln|\cos\theta_s| - \ln(R_s^2)}{\ln|\cos\theta| - \ln R^2} + K_4 \tag{9}$$

where $R_s$ is the reference distance and $I_s$ is the corrected intensity value, $K_4 = K_3 + \tau$.

In this study, the reference distance was set as the average flight altitude of the UAV of 60 m, and the reference incident angle was set to 0°. Since 0 has no logarithm, when the incident angle $\theta$ equals 90°, $\cos\theta$ is set to $1^{-150}$. The laser ranging value R is $(flight\ height - z)/\cos\theta$ with flight height equal to 60; $K_1$ and $K_2$ are model coefficients obtained by fitting the least square method using $I$, R, $\theta$.

The number of points in an ITC point cloud varies greatly for the same tree species due to the different ITC shapes. This can affect the intensity frequency. To reduce the effect of the number of points in an ITC point cloud on the intensity frequency, the frequency value associated to the intensity value is multiplied by a weight. This weight is computed using the inverse distance weighting (IDW) according to Equation (10).

$$\varphi_i = \frac{\frac{1}{p_i}}{\sum_{i=1}^{n}\frac{1}{p_i}} \tag{10}$$

where $\varphi_i$ represents the weight of the number of the i-th ITC point cloud, $p_i$ represents the number of points in the i-th ITC point cloud, and n is the number of ITCs.

To analyze the influence of canopy intensity frequency on tree species classification accuracy under different point cloud densities, this study set three sampling rates of 80%, 50%, and 30% to resample the D1 data set by the random thinning method [42]; this can be helpful in estimating the most adequate number of features, and thus reducing the complexity and decreasing the computational time. The point density of the three resampled point clouds, D2, D3, and D4 as well as D1 are shown in Table 2.

**Table 2.** Density of point clouds.

| Data Set Name | Sampling Rate | Mean Point Density |
|:---:|:---:|:---:|
| D1 | 100% | 230 points/m$^2$ |
| D2 | 80% | 184 points/m$^2$ |
| D3 | 50% | 115 points/m$^2$ |
| D4 | 30% | 69 Points/m$^2$ |

### 2.5. Intensity Frequency Feature Calculation and Difference Analysis

The intensity frequency feature process obtained from the ITC point cloud is shown in Figure 3. First, the already-corrected intensity of the ITC point cloud is assessed to obtain the intensity frequencies. Then, an intensity frequency value is assigned to an intensity frequency feature. For example, suppose an ITC point cloud of G.B with intensity values from 3 to 150, where the number of times the point with intensity 3 appears in this ITC point cloud is 10, is said to be $IF_3 = 10$. After multiplying by $\varphi_i$ and rounding, $IF_3$ is considered as an intensity frequency feature. This study acquired 147 intensity frequency features from $IF_3$ to $IF_{150}$.

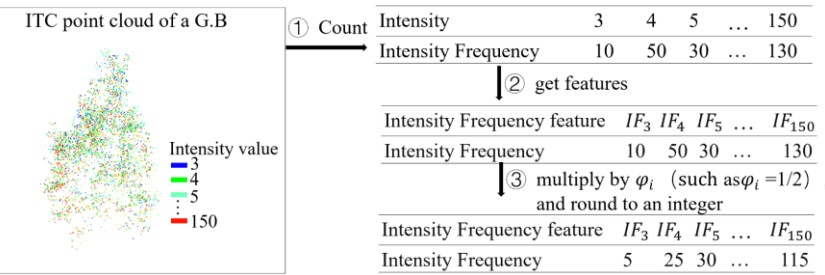

**Figure 3.** Intensity frequency feature acquisition schematic.

As shown in Figure 4, the Savitzky-Golay (S-G) filter (Savitzky, 1964) was selected for data stream smoothing [43] to enhance the characteristics of each ITC intensity frequency curve. The S-G filter is a filtering method based on a local polynomial least squares fit. The most important feature of this filter is that the shape and width of the signal can be kept constant while filtering the noise. This process is implemented by calling 'scipy. signal' in Python, where the smoothed window length M is taken as 51, the polynomial order is set to 3, and the fill signal of the filter is set to the nearest mode.

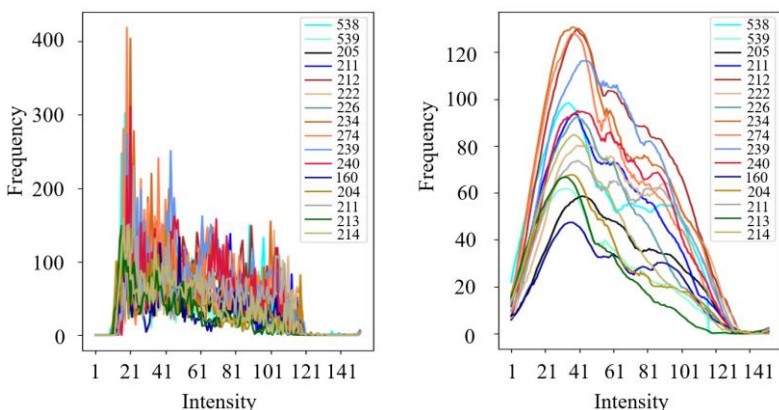

**Figure 4.** A before-and-after diagram of the Savitzky-Golay (S-G) filter; the numbers in the legend are the serial numbers of the 16 ITC point clouds with the tree species G.B; the line colors indicate the intensity frequencies of the different number of ITC point clouds.

To test the hypothesis that intensity frequencies differ significantly among tree species and not significantly among the same tree species, a nonparametric Kruskal-Wallis (KW) test [44] was used, followed by a comparison of the individual tree species intensity frequency features using the mean rank (MR). Post hoc analyses for KW and MR were performed using SPSS (IBM v.26).

### 2.6. Random Forest and Tree Species Classification

A random forest (RF) is a combination model with many decision trees [45]. The classification process is as follows: bootstrapping is used to form N bootstrap samples, a classification tree model is established for each bootstrap sample, and the sample corresponds to all the training data and test data of the classification tree model. Finally, the model of N classification tree results is taken as the final classification result.

For training sample data with a large number of differences, a sample with a dominant number will have a greater impact on the random forest. The attribute weights produced by an RF on this kind of data are not credible. To maintain a relatively balanced number of tree species, 446 samples were taken in the study to use in the classification of the following tree species: *Ginkgo Biloba* (G.B: n = 86), *Michelia Figo* (M.F: n = 52), *Cinnamomum camphora* (C.C: n = 64), *Celtis Sinensis Pers* (C.S: n = 27), *Acer Buergerianum Miq* (A.B: n = 71), *Salix Babylonica* (S.B: n = 78), *Glyptostrobuspensilis* (G.L: n = 41), and *Magnolia grandiflora*

(M.G: n = 27). The training and testing data sets were divided into 313 and 133 samples at a ratio of 7:3. The number of decision trees was set to 500 to ensure that each sample was classified more than once. A total of 147 intensity features were input into the RF model. The RF was performed in Python.

### 2.7. Tree Species Classification Process

In summary, the flow chart of this research is shown in Figure 5. First, the raw data are decoded to LAS format point clouds. Then, the LAS point clouds are denoised, normalized, and segmented to the ITC point clouds. ITC point clouds are labeled as tree species using field data in ArcGIS. Then, intensity correction data set (100%) were resampled as other three density data sets and the 147 intensity frequency features of each ITC of each data set are obtained. Finally, four data sets are classified with RF and then assessed.

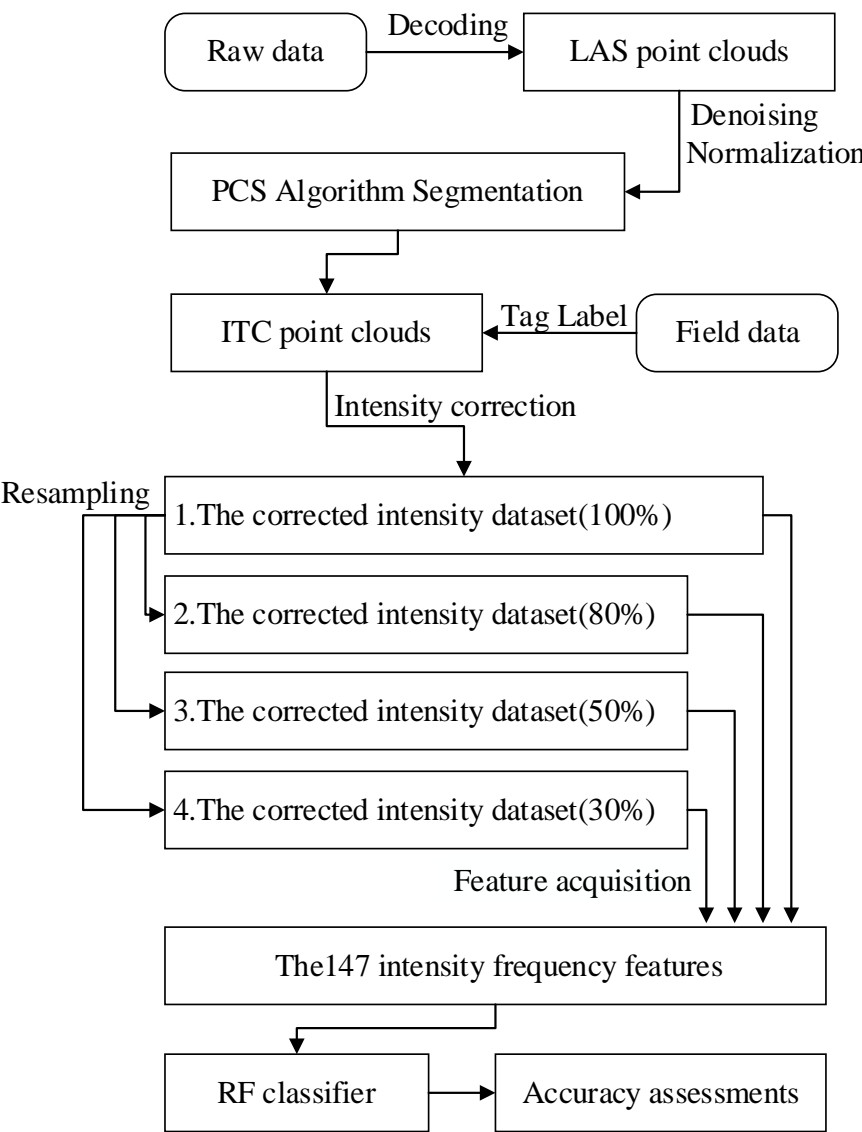

**Figure 5.** Flow chart.

### 2.8. Accuracy Evaluation

The measures of the per-tree species classification include producer accuracy (PA, Equation (11)), user accuracy (UA, Equation (12)), commission errors (CE, Equation (13)),

and omission errors (OE, Equation (14)). The measure used for the all-tree species classification is the overall accuracy (OA, Equation (15)).

$$PA = \frac{N_{rr}}{N_{rr} + N_{rn}} \tag{11}$$

$$UA = \frac{N_{rr}}{N_{rr} + N_{nr}} \tag{12}$$

$$CE = \frac{N_{nr}}{N_{rr} + N_{nr}} = 1 - UA \tag{13}$$

$$OE = \frac{N_{rn}}{N_{rr} + N_{rn}} = 1 - PA \tag{14}$$

$$OA = \frac{N_{corr}}{N_{total}} = \frac{N_{rr}}{N_{rr} + N_{nr} + N_{rn}} \tag{15}$$

where for a given tree species, $N_{rr}$ represents the number of correctly classified samples, $N_{rn}$ represents the number of samples of that species wrongly classified, and $N_{nr}$ represents the number of samples of another tree species wrongly classified as the given species. The OA is the total number of samples of any tree species classified correctly ($N_{corr}$) divided by the total number of validation samples ($N_{total}$).

## 3. Results

### 3.1. ITC Extraction Results

Figure 6a gives an example of G.B point clouds and Figure 6b shows the ITC detected with the PCS algorithm. As seen in Figure 6b, the PCS algorithm is able to detect ITC successfully. A total of 987 trees were correctly detected within the eight tree species using the PCS algorithm, accounting for 89.2% of the total trees surveyed; 123 (11.8%) ITCs were not detected, and the number of ITCs that were not present by incorrect detection was 139 (13.6%). Table 3 shows the ITC detection precision relative to the eight tree species. As can be seen from Table 3, the detection accuracy of ITC is promising, with M.F having the highest detection accuracy of 93.4%, followed by C.S with an accuracy of 91.1%, and G.L with a relatively low detection accuracy of 82.4%.

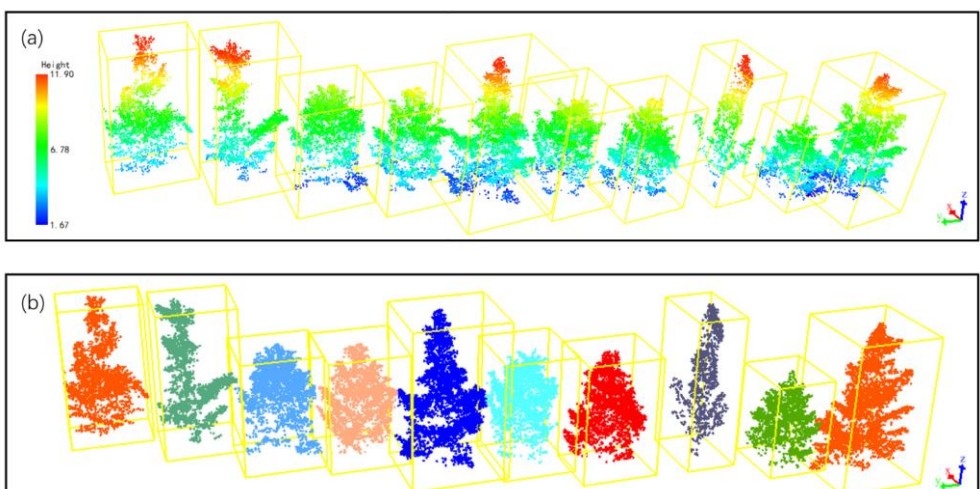

**Figure 6.** (**a**) Point clouds of G.B numbered 1–10; (**b**) segmentation results using the PCS algorithm (different colors correspond to different ITCs).

**Table 3.** ITC segmentation accuracy assessment. The darker color indicates higher accuracy.

| Tree Species | Nt | No | Nc | R (%) | P (%) | F (%) |
|:---:|:---:|:---:|:---:|:---:|:---:|:---:|
| M.F | 85 | 6 | 10 | 93.4% | 89.5% | 91.4% |
| C.C | 167 | 24 | 19 | 87.4% | 89.8% | 88.6% |
| C.S | 51 | 5 | 7 | 91.1% | 87.9% | 89.5% |
| S.B | 81 | 10 | 8 | 89.0% | 91.0% | 90.0% |
| A.B | 62 | 11 | 13 | 84.9% | 82.7% | 83.8% |
| G.B | 435 | 54 | 67 | 89.0% | 86.7% | 87.8% |
| M.G | 39 | 8 | 11 | 83.0% | 78.0% | 80.4% |
| G.L | 56 | 12 | 15 | 82.4% | 78.9% | 80.6% |

### 3.2. Intensity Correction Results

The average intensity frequencies of each tree species extracted based on Equations (4)–(10) are shown in Figure 7. The average intensity frequency of a tree species is the average intensity frequency of all ITC point clouds of this species. Figure 7a shows that the average ITC intensity frequency of the eight species is different, among which C.S is the highest and G.B is the lowest. These differences provide the conditions for the classification of tree species.

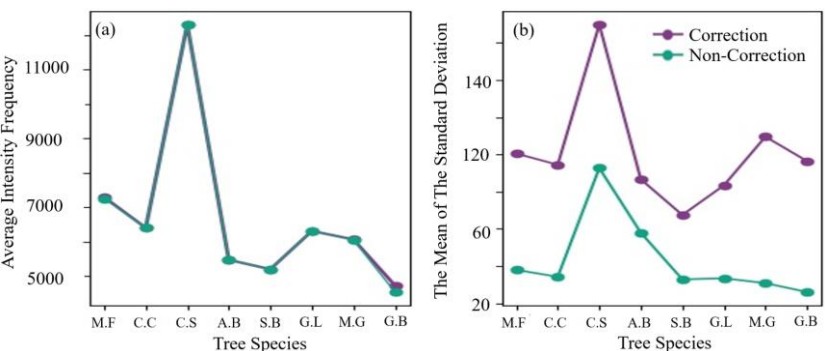

**Figure 7.** (**a**) Average intensity frequency of each species; (**b**) mean standard deviation of the intensity frequency of each species.

Figure 7b shows the standard deviation of each tree species' average intensity frequency after intensity correction. The mean standard deviation of the intensity of a tree species is the average of the standard deviation of the intensity values of all ITC point clouds for this species. The average standard deviation after correction is reduced relative to before correction, and this is likely to be because of the logarithmic correction model. This model reduces the error in intensity caused by the laser beam range and incident angle, and it produces an intensity value that is mainly a function of the target reflectance characteristics.

### 3.3. Results of Intensity Frequency of Different Species

Figure 8a shows a diagram of point clouds with sampling rates of 100%, 80%, 50%, and 30% for the eight tree species. As seen in Figure 9a, from top to bottom, the point cloud gradually becomes sparse as the sampling rate decreases, but the canopy outline and key points remain relatively intact, which is expected because the segmentation was carried out with the full density point cloud D1. From left to right, the canopy shape, canopy point cloud imaging, and point cloud density of different tree species vary. For example, the point cloud of G.B is relatively sparse compared with the other seven species, which may result from the fact that G.B itself has smaller leaves and a large gap between leaves and does not form a tightly expanded canopy shape, thus resulting in a smaller number of point clouds. This shows that point clouds can reflect the differences in leaf shape and canopy structures of different trees.

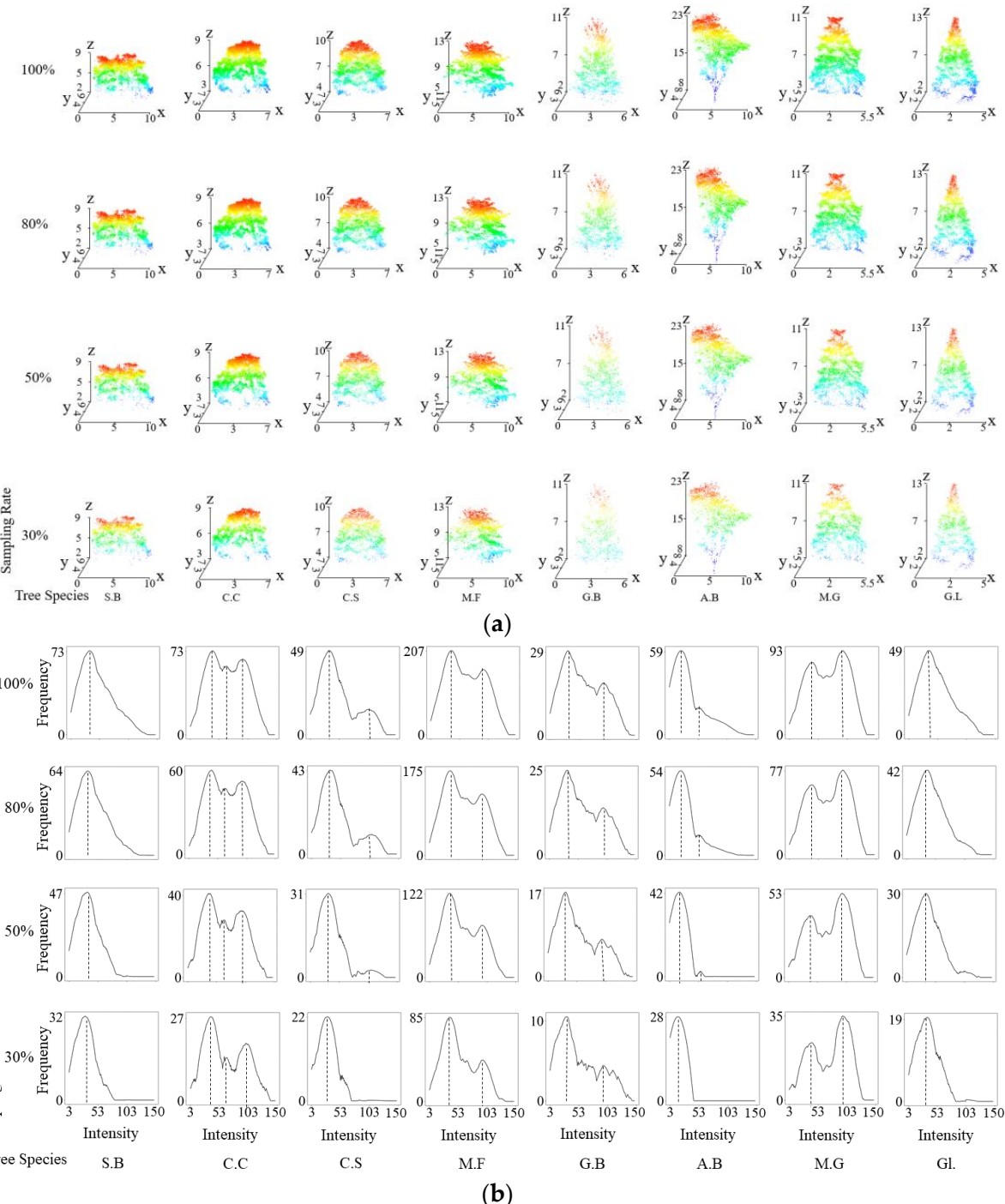

**Figure 8.** (**a**) Pointclouds of different densities of the eight tree species; the vertical coordinate of (**b**) is the maximum intensity frequency value among all the detected ITC per species. The 8 different colored bars above the same IF feature indicate the MR of the 8 tree species.

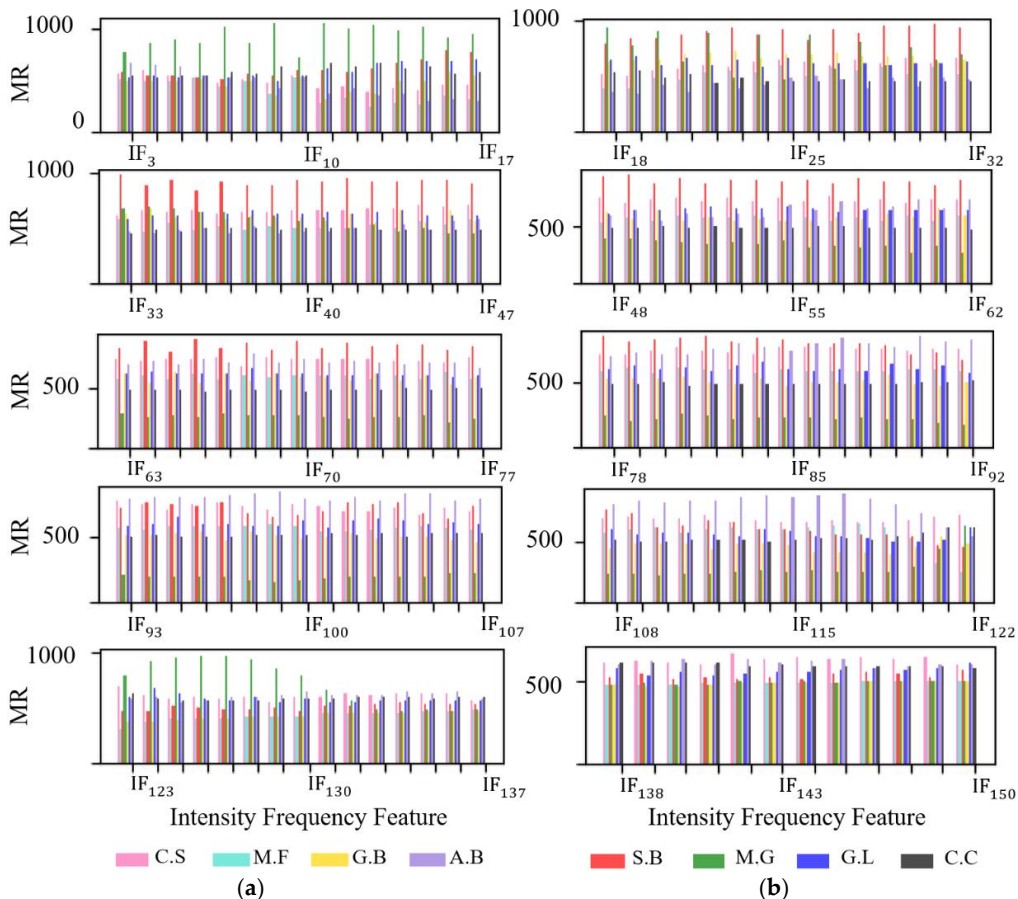

**Figure 9.** (**a**,**b**) Mean rank (MR) histogram of 147 intensity features of 8 tree species. Different tree species are indicated by different colors. the values of MR represent the differences in IF features among the eight tree species.

Figure 8b shows the intensity frequency curves of point clouds at different densities of the eight tree species. From left to right, the characteristics of the intensity frequency curve of each tree species are different, such as peak value, wave peak number, and maximum frequency, providing conditions for the classification of tree species using intensity frequency. For a given tree species, the shape of the curve is similar as the sampling rate decreases, but the troughs show an overall decreasing trend. This may be due to the fact that the trough itself is small, and the relative number decreases more after random sampling, so the curve shows a downward trend of the trough. The decrease in sampling rate, i.e., the decrease in point cloud density, causes the curve to lose some details, which may lead to a decrease in the correct classification rate of tree species and may lead to the misclassification of tree species. As an example, when the sampling rate is 30% for C.S, the second wave of its frequency curve disappears. The shape and peak of the intensity frequency curve are similar to those of S.B and G.L, which may lead to C.S being misclassified as S.B and G.L. This further indicates that the ability of point cloud intensity frequencies to represent differences in tree species is influenced by point cloud density.

### 3.4. Intensity Frequency Difference Analysis Results

The results of the intensity frequency difference analysis are shown in Table 4. Table 4 shows that the p values of the Kruskal–Wallis H-test of the intraspecies intensity frequencies of the eight trees are all greater than 0.05, while the interspecies p values are all less than 0.01, indicating that the intraspecies differences in the intensity frequency are not significant, but the interspecies intensity frequency differences are significant. This provides a theoretical basis for distinguishing eight tree species based on intensity frequency. In

addition, Table 4 also shows the impact of intensity frequency sampling on tree species classification differences. The analysis shows that the interspecies statistic H(K) decreases gradually with the decrease in sampling rate, while the intraspecies p and H(K) remain unchanged. This may indicate that the overall statistical variation of interspecies intensity frequencies decreases as the number of points in the point clouds decreases; the statistical variation of intraspecies intensity frequencies remains constant with some stability.

**Table 4.** Kruskal-Wallis H test results of intraspecies and interspecies.

| Sampling Rate | | 100% | | 80% | | 50% | | 30% | |
|---|---|---|---|---|---|---|---|---|---|
| | | p | H(k) | p | H(k) | p | H(k) | p | H(k) |
| Intraspecies | M.F | 0.474 | 51 | 0.474 | 51 | 0.474 | 51 | 0.474 | 51 |
| | C.C | 0.476 | 63 | 0.476 | 63 | 0.476 | 63 | 0.476 | 63 |
| | C.S | 0.462 | 25 | 0.462 | 25 | 0.462 | 25 | 0.462 | 25 |
| | A.B | 0.478 | 70 | 0.478 | 70 | 0.478 | 70 | 0.478 | 70 |
| | S.B | 0.479 | 77 | 0.479 | 77 | 0.479 | 77 | 0.479 | 77 |
| | G.L | 0.470 | 40 | 0.470 | 40 | 0.470 | 40 | 0.470 | 40 |
| | M.G | 0.463 | 26 | 0.463 | 26 | 0.463 | 26 | 0.463 | 26 |
| | G.B | 0.480 | 85 | 0.480 | 85 | 0.480 | 85 | 0.480 | 85 |
| Interspecies | | <0.01 | 181.590 | <0.01 | 133.241 | <0.01 | 106.956 | <0.01 | 87.638 |

Figure 9 shows the MR histogram of each tree species under 147 features. There are eight bars above each feature, indicating eight tree species. The height of the bars represents the mean rank (MR) of that tree species under this feature. Statistically, if the MR differs significantly, the population to which the two groups of samples belong is considered significantly different. By looking at the bars with significant differences in height, it is possible to initially see which tree species are more different from other species in terms of features. In this study, 4% of the mean MR was used as the threshold [46] to screen the most relevant features of various tree species, as summarized in Table 5.

**Table 5.** Most relevant intensity frequency features for each species selected by MR in D1. 'IF$_a$–IF$_b$' means that all from IF$_a$ to IF$_b$ were selected.

| Tree Species | Important Intensity Frequency Features in D1 |
|---|---|
| M.F | IF$_3$, IF$_8$−IF$_{13}$, IF$_{18}$, IF$_{21}$, IF$_{45}$, IF$_{41}$−IF$_{64}$, IF$_{66}$−IF$_{115}$, IF$_{117}$−IF$_{121}$, IF$_{132}$, IF$_{133}$, IF$_{140}$, IF$_{142}$−IF$_{144}$ |
| C.C | IF$_4$, IF$_6$, IF$_8$−IF$_{16}$, IF$_{44}$, IF$_{49}$, IF$_{50}$, IF$_{55}$, IF$_{60}$, IF$_{63}$, IF$_{64}$, IF$_{70}$, IF$_{73}$, IF$_{74}$, IF$_{76}$, IF$_{101}$, IF$_{106}$, IF$_{110}$, IF$_{113}$, IF$_{116}$, IF$_{118}$−IF$_{131}$ |
| C.S | IF$_{12}$−IF$_{15}$, IF$_{18}$−IF$_{53}$, IF$_{55}$−IF$_{57}$, IF$_{59}$, IF$_{61}$−IF$_{63}$, IF$_{65}$−IF$_{68}$, IF$_{70}$−IF$_{84}$, IF$_{95}$, IF$_{97}$−IF$_{107}$, IF$_{109}$, IF$_{112}$, IF$_{114}$, IF$_{116}$, IF$_{119}$−IF$_{121}$ |
| A.B | IF$_3$−IF$_5$, IF$_7$, IF$_9$−IF$_{19}$, IF$_{25}$, IF$_{27}$, IF$_{29}$−IF$_{32}$, IF$_{36}$, IF$_{43}$−IF$_{118}$, IF$_{120}$−IF$_{125}$, IF$_{127}$, IF$_{129}$ |
| S.B | IF$_4$, IF$_6$, IF$_8$−IF$_{10}$, IF$_{15}$, IF$_{18}$−IF$_{20}$, IF$_{22}$−IF$_{27}$, IF$_{30}$, IF$_{35}$, IF$_{39}$, IF$_{49}$, IF$_{54}$, IF$_{57}$, IF$_{64}$, IF$_{66}$, IF$_{70}$, IF$_{104}$, IF$_{105}$, IF$_{109}$, IF$_{113}$−IF$_{116}$, IF$_{120}$−IF$_{133}$ |
| G.L | IF$_{11}$−IF$_{14}$, IF$_{16}$, IF$_{24}$, IF$_{25}$, IF$_{27}$, IF$_{29}$, IF$_{34}$, IF$_{44}$, IF$_{47}$, IF$_{54}$, IF$_{60}$, IF$_{66}$, IF$_{76}$, IF$_{78}$, IF$_{97}$−IF$_{99}$, IF$_{103}$, IF$_{106}$, IF$_{109}$, IF$_{111}$, IF$_{119}$, IF$_{121}$−IF$_{123}$, IF$_{131}$, IF$_{134}$, IF$_{135}$, IF$_{143}$, IF$_{145}$ |
| M.G | IF$_4$, IF$_5$, IF$_9$, IF$_{11}$−IF$_{24}$, IF$_{28}$, IF$_{30}$, IF$_{34}$, IF$_{37}$, IF$_{39}$, IF$_{46}$, IF$_{49}$, IF$_{55}$, IF$_{57}$, IF$_{60}$, IF$_{62}$−IF$_{86}$, IF$_{88}$, IF$_{89}$, IF$_{91}$−IF$_{96}$, IF$_{100}$−IF$_{103}$, IF$_{106}$−IF$_{109}$, IF$_{113}$−IF$_{121}$, IF$_{126}$, IF$_{129}$, IF$_{137}$, IF$_{140}$, IF$_{142}$, IF$_{145}$, IF$_{146}$ |
| G.B | IF$_4$, IF$_6$, IF$_9$, IF$_{11}$−IF$_{13}$, IF$_{19}$, IF$_{21}$−IF$_{23}$, IF$_{25}$−IF$_{30}$, IF$_{120}$−IF$_{125}$, IF$_{146}$ |

### 3.5. Screening Results of Important Random Forest Features

Figure 10 shows the top 10 features with the most relevant importance for tree classification of the eight tree species using the RF model in D1. As seen in the figure, the top 10 most relevant features of the eight tree species accounted for more than 34% of the total 147 features. Among them, the most relevant of the top 10 features of A.B accounted for 88%.

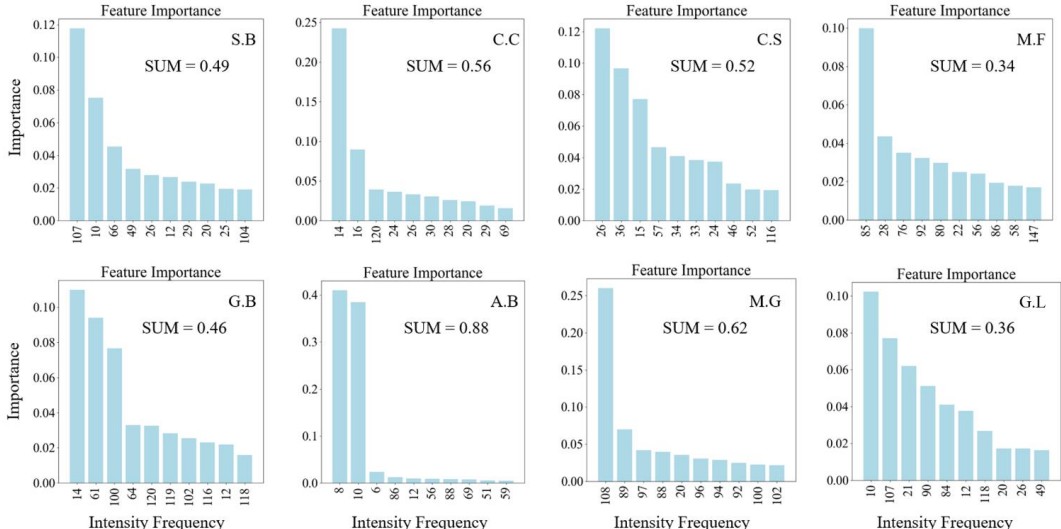

**Figure 10.** Important intensity frequency features for each species selected by RF in D1. The subfigure shows the most relevant features of a tree species. 'SUM' equals the sum of the importance of the most relevant features calculated by RF.

Comparing Figure 10 and Table 5, it can be seen that the priori feature selected by MR covers the top 10 posterior features selected by RF. This can provide a reference for selection of the most relevant features using the MR before classification in the future.

For M.F and G.L, the proportions of the top 10 features are 34% and 36%, respectively, consistent with the classification results in Section 3.6. A.B is stable and has the high UA and PA among all species, indicating that the top 10 features are relevant, the tree species have prominent features, and the classification accuracy is high.

### 3.6. Tree Species Classification Results

The results of tree species classification with an RF model based on 147 intensity frequency features are shown in Table 6. As seen in Table 6, the UA of tree species classification can range from 100% down to 82.6%. The PAs of three tree species reached more than 90%, three species reached more than 85%, and two species had PAs of 75.0%. The lowest CE is 0%, and the lowest OE is 4.8%. AU is higher than PU, and CE is lower than OE. This shows that intensity frequency can provide effective information for tree species classification. The accuracy of tree species classification using intensity frequency features is high.

**Table 6.** Tree species classification results (D1, sampling rate 100%), where Num refers to the number of these species in the validation set. The sum of correct classification trees (SCI) is the number of correctly classified trees in a validation set of 133.

| Name | M.F | C.C | C.S | A.B | S.B | Gl. | M.G | G.B | UA (%) | CE (%) |
|---|---|---|---|---|---|---|---|---|---|---|
| Num. | n = 16 | n = 19 | n = 8 | n = 21 | n = 23 | n = 12 | n = 8 | n = 26 | | |
| M.F | 12 | 0 | 1 | 0 | 0 | 0 | 0 | 1 | 85.7 | 14.3 |
| C.C | 1 | 19 | 1 | 0 | 2 | 0 | 2 | 0 | 82.6 | 17.4 |
| C.S | 0 | 0 | 7 | 0 | 0 | 0 | 0 | 0 | 100.0 | 0 |
| A.B | 0 | 0 | 0 | 20 | 0 | 0 | 0 | 0 | 100.0 | 0 |
| S.B | 3 | 0 | 2 | 0 | 21 | 0 | 0 | 0 | 100.0 | 0 |
| Gl. | 0 | 0 | 0 | 0 | 0 | 9 | 0 | 1 | 90.0 | 10.0 |
| M.G | 0 | 0 | 0 | 0 | 0 | 0 | 6 | 0 | 100 | 0 |
| G.B | 1 | 0 | 0 | 0 | 0 | 2 | 1 | 23 | 85.1 | 14.9 |
| PA (%) | 85.7 | 95.0 | 87.5 | 95.2 | 91.3 | 75.0 | 75.0 | 88.5 | OA: 86.7% | |
| OE (%) | 14.3 | 5.0 | 12.5 | 4.8 | 8.7 | 25.0 | 25.0 | 11.5 | SCI: 117 | |

As shown in Tables 6–9, the sum of correct classification trees (SCI) decreases as the sampling rate decreases: SCI equals 117 at D1 data set, 97 in D2, 74 in D3, and 65 in D4. The largest decrease in SCI occurs between the D2 and D3. This is probably because the sampling rate decreases the most (by 30%) from D2 to D3. The number of point clouds is reduced more, and more key points that reflect the differences in intensity frequencies of each tree species may be lost, leading to a decrease in classification accuracy. Therefore, the SCI at an 80% sampling rate differs the most from 50% relative to the other two cases, indicating that the point cloud density positively correlates with the sum of correct classification trees (SCIs) in this study.

**Table 7.** Tree species classification results (D2, sampling rate 80%), where Num refers to the number of these species in the validation set. The sum of correct classification trees (SCI) is the number of correctly classified trees in a validation set of 133.

| Name | M.F | C.C | C.S | A.B | S.B | Gl. | M.G | G.B | UA (%) | CE |
|------|-----|-----|-----|-----|-----|-----|-----|-----|--------|------|
| Num. | n = 16 | n = 19 | n = 8 | n = 21 | n = 23 | n = 12 | n = 8 | n = 26 | | (%) |
| M.F | 10 | 1 | 1 | 0 | 0 | 1 | 0 | 0 | 76.9 | 23.1 |
| C.C | 1 | 15 | 0 | 0 | 0 | 0 | 0 | 0 | 93.3 | 6.7 |
| C.S | 0 | 1 | 4 | 0 | 0 | 0 | 0 | 0 | 80.0 | 20.0 |
| A.B | 0 | 0 | 0 | 19 | 0 | 0 | 0 | 1 | 95.0 | 5.0 |
| S.B | 1 | 0 | 0 | 0 | 17 | 1 | 0 | 0 | 88.0 | 12.0 |
| Gl. | 1 | 0 | 0 | 0 | 0 | 7 | 0 | 3 | 63.6 | 56.4 |
| M.G | 0 | 1 | 0 | 0 | 0 | 0 | 6 | 0 | 85.7 | 14.3 |
| G.B | 0 | 0 | 0 | 0 | 0 | 1 | 0 | 19 | 95.0 | 5.0 |
| PA (%) | 62.5 | 78.9 | 50.0 | 90.5 | 95.7 | 58.3 | 75.0 | 84.6 | **OA: 87.4%** | |
| OE (%) | 37.5 | 21.1 | 50.0 | 9.5 | 4.3 | 41.7 | 25.0 | 15.4 | **SCI: 97** | |

**Table 8.** Tree species classification results (D3, sampling rate 50%), where Num refers to the number of these species in the validation set. The sum of correct classification trees (SCI) is the number of correctly classified trees in a validation set of 133.

| Name | M.F | C.C | C.S | A.B | S.B | Gl. | M.G | G.B | UA (%) | CE |
|------|-----|-----|-----|-----|-----|-----|-----|-----|--------|------|
| Num. | n = 16 | n = 19 | n = 8 | n = 21 | n = 23 | n = 12 | n = 8 | n = 26 | | (%) |
| M.F | 7 | 0 | 0 | 0 | 0 | 0 | 0 | 1 | 87.5 | 12.5 |
| C.C | 3 | 12 | 1 | 0 | 0 | 0 | 0 | 2 | 66.7 | 33.3 |
| C.S | 1 | 0 | 3 | 0 | 0 | 0 | 0 | 0 | 75.0 | 25.0 |
| A.B | 0 | 0 | 0 | 20 | 0 | 0 | 0 | 1 | 100.0 | 0 |
| S.B | 1 | 0 | 0 | 0 | 8 | 0 | 0 | 0 | 88.0 | 12.0 |
| Gl. | 1 | 0 | 0 | 0 | 0 | 3 | 0 | 0 | 75.0 | 25.0 |
| M.G | 0 | 0 | 0 | 0 | 0 | 0 | 4 | 1 | 80.0 | 20.0 |
| G.B | 0 | 1 | 1 | 0 | 0 | 1 | 0 | 17 | 85.0 | 15.0 |
| PA (%) | 62.5 | 63.2 | 37.5 | 95.2 | 34.7 | 25.0 | 50.0 | 65.4 | **OA: 84.1%** | |
| OE (%) | 37.5 | 36.8 | 62.5 | 4.8 | 65.3 | 75.0 | 50.0 | 34.6 | **SCI: 74** | |

**Table 9.** Classification results of tree species (D4, sample rate 30%), where Num refers to the number of these species in the validation set. The sum of correct classification trees (SCI) is the number of correctly classified trees in a validation set of 133.

| Name | M.F | C.C | C.S | A.B | S.B | Gl. | M.G | G.B | UA (%) | CE |
|---|---|---|---|---|---|---|---|---|---|---|
| Num. | n = 16 | n = 19 | n = 8 | n = 21 | n = 23 | n = 12 | n = 8 | n = 26 | | (%) |
| M.F | 3 | 1 | 0 | 0 | 0 | 0 | 0 | 0 | 75.0 | 25 |
| C.C | 0 | 7 | 0 | 0 | 0 | 0 | 0 | 2 | 77.8 | 22.1 |
| C.S | 0 | 2 | 4 | 0 | 0 | 0 | 0 | 0 | 66.7 | 33.3 |
| A.B | 0 | 0 | 0 | 19 | 0 | 0 | 0 | 1 | 100.0 | 0 |
| S.B | 0 | 0 | 2 | 0 | 11 | 0 | 0 | 0 | 84.6 | 15.4 |
| Gl. | 1 | 0 | 0 | 0 | 0 | 1 | 0 | 0 | 50.0 | 50.0 |
| M.G | 0 | 1 | 0 | 0 | 0 | 0 | 4 | 0 | 80.0 | 20.0 |
| G.B | 1 | 0 | 0 | 0 | 0 | 0 | 0 | 16 | 94.1 | 5.9 |
| **PA (%)** | 18.8 | 63.2 | 37.5 | 90.4 | 47.8 | 25.0 | 8.3 | 61.5 | **OA: 85.5%** | |
| **OE (%)** | 81.2 | 36.8 | 62.5 | 9.6 | 52.2 | 75.0 | 91.7 | 38.5 | **SCI: 65** | |

UA remains relatively stable across the four sampling rates in the data set. The reason may be because most of the classified samples after random sampling still provide the critical information required by RF for successful classification. The UA remains relatively stable, which is consistent with the implication shown in Figure 8 that the intensity frequency curve retains its basic shape after resampling, but the details change. This indicates that the intensity frequency feature can maintain a relatively robust UA at different point cloud densities in this study. The PA decreases with decreasing sampling rate, potentially because as the sampling rate decreases, some of the intensity frequency features can no longer provide the information needed for RF to classify tree species, leading to an increase in the number of true samples classified as other ($N_{rn}$).

Finally, the overall accuracy (OA) is above 84% for all four data sets, with the highest being the D2 data set (OA = 87.4%), slightly higher than that of the D1 data set. The reason could be that as the sampling rate decreases, some intensity frequency features of ITC, although able to be classified by RF at a 100% sampling rate, have a high error rate. RF can no longer classify them at an 80% sampling rate, which leads to an increase in $N_{rn}$ since the overall accuracy = $(N_{rr})/(N_{rr} + N_{nr} + N_{rn})$. While the $N_{rr}$ at an 80% sampling rate decreases, the $N_{rn}$ also decreases. The numerator decreases less than the denominator, resulting in a slight increase in overall precision.

In summary, the intensity frequency features of four data set have promising results on UA, PA, CE, OE, OA, and SCI. This indicates that the intensity frequency features of the ITC point cloud of eight tree species obtained from this study can accurately perform tree species classification.

## 4. Discussion

The results show that the LiDAR intensity frequency, combined with the PCS and RF algorithms, can achieve high-precision classification of eight urban forest tree species in the study area.

Previous studies have shown that intensity data can realize surface classification and object detection [30,47]. The target water content significantly influences the intensity [48,49]. Both the reflectivity and the structural characteristics of ITC would influence the intensity of the intensity of point clouds acquired by UAV-LiDAR [49]. Different tree species may have different water contents, canopy structures, and physiological and biochemical characteristics of leaves [50]. Therefore, the intensity features of LiDAR have promising potential for tree species classification.

LiDAR intensity data have some advantages for tree species classification over passive remote sensing sensor data such as multispectral and hyperspectral samples [25,51,52]. First, the measurement of LiDAR intensity has the advantage of being independent of

external lighting conditions [31], so LiDAR data are not affected by variable shadows and natural light, and the positional accuracy of point clouds is, in general, more reliable when compared to that of point clouds produced with aerial images [53]. Second, although passive remote sensing imaging data can be reversed to generate point clouds, in vegetated areas, such point clouds have limited penetration and ground points. The LiDAR point cloud will contain background information, contrary to that produced with images, allowing to the structuring of information. In addition, the incidence angle and height information provided by LiDAR is brought into some intensity correction models. This can reduce the intensity data differences caused by distance and angle of incidence to some extent so that the intensity contains the difference in target reflectivity [30]. Finally, LiDAR can provide both structural and intensity information, which has the advantages of requiring less storage space, and less complex and time-consuming data processing compared to hyperspectral and multispectral data [54,55]. There are also advantages when using intensity frequencies versus individual intensities for tree species classification. A single intensity can only reflect the characteristics of one point of the whole tree canopy and carries limited information. If there is a point cloud of reflections from other features, it may cause significant interference with the classification results. In contrast, intensity frequency is the collection of all intensities of the whole canopy, i.e., carrying intensity information and having a small degree of anti-interference ability. The intensity frequencies can provide pseudo-waveform features similar to the spectral reflectance curves. The intensity frequency feature of pseudo-waveform could provide more tree species differences information than single-valued features.

Current studies using intensity for tree species classification include converting intensity into raster images and then combining other features for species classification [23,47,56,57] or extracting the mean and standard deviation of intensity from manually depicted tree canopies for species classification [58]. The advantage of this approach is in preserving the spatial distribution characteristics of the intensity data, while the disadvantage is that the unit raster intensity value is the sum of the intensity of the points that are projected into the pixel or the standard deviation of those intensity values, which may somewhat lose the structure of the target object to which the intensity responds. The other method involves constructing vegetation indices for tree species classification using multiple LiDAR sensors with different wavelengths; for example, Yu et al. (2017) classified pine, spruce, and birch trees based on point cloud single-channel intensity, multichannel intensity, and combined data of three channels with a maximum accuracy of 85.9% [25]. The advantage of this method is that it enhances the differences between different tree species by constructing vegetation indices, thus making it easier to classify tree species. The disadvantage is that acquiring data from different channels requires multiple repetitive operations with multiple sensors, which is time-consuming and expensive. Additionally, the vegetation indices that can induce the response of the intensity curves of each tree species are not yet clear, which is also a worthwhile research direction in the future.

Although we obtained a qualitative result showing that the intensity frequencies of the eight tree species did not differ significantly for intraspecies or interspecies observations by the KW test, we did not perform a more detailed quantitative analysis for each tree species. Future research could be conducted for a quantitative analysis of the differences in intensity frequencies among tree species.

In Table 4, the PU and UA of some species need further improvement, such as the PA of both G.L and M.G being 75%. The main reason for the low PA accuracy of these two species may be that the training samples of these two species are few, and the RF model cannot thoroughly learn the intensity frequency characteristics of these two species. As a result, the intensity frequency does not possess the tree's complete canopy characteristics, resulting in a decrease in the PA. At the same time, the classification error of tree species in this study may also be derived from two aspects. First, the difference in canopy structure information and the crown shapes of different individual plants of the same tree species are often similar but different. Second, the physiological and biochemical status of target

leaves may also bring about tree species classification errors. Different crown properties affect the distribution of laser echoes within and on the surface of the tree crowns [59], thus affecting the intensity frequency. More species lead to more diversity in leaf shape, size, and reflectance, as well as in tree structure, resulting in more difficult classification tasks. In this case, up to eight species are classified using only the intensity signatures from a near-ground flight (60 m) UAV, with an overall accuracy of over 86%. To our knowledge, such results have not been reported in the scientific literature.

This study was based on ITC point clouds for point cloud thinning and analyzed the effect of density on the classification accuracy of tree species. This approach followed the principle of controlling a single variable for the comparison test [60]. Density as the only dependent variable when studying the effect of density on classification results. The effect of canopy segmentation differences on classification accuracy was excluded, because point clouds with different densities would lead to different results of PCS segmentation of canopies. Therefore, the conclusion related to the influence of point cloud density on classification accuracy in this study does not apply to point clouds of different densities acquired at different heights. This is also a direction worth investigating. There are some limitations in this study. First is the limitation of ITC segmentation; the method of this study is applicable to point clouds so that ITC can be segmented accurately. However, in complex terrain, complex forest conditions, and an automatic acquisition of segmented canopy, unavoidable random or systematic errors may arise, which leads to elevated classification errors [61]. The second limitation is that the topography of this study area is relatively flat. If the topography is complex, the data noise of each canopy will be greater, and the classification will be less reliable. In addition, during intensity correction, we treat the observation error $\tau$ as a constant, which may also cause errors in the intensity correction. Finally, there is the diversity of tree canopies. Theoretically, different tree species have different spectral reflectance properties, and the same tree species has similar spectral reflectance properties [62]. However, the diversity of canopies caused by the variation in canopy structure, leaf shape, and reflectance with age increases the intraspecific variability and classification error probability [25]. Therefore, further research is necessary to follow up on whether stratification by tree height or age can improve classification performance. In addition, random forest models, trained using a limited number of samples from a specific region, are difficult to apply on a broader scale without additional training samples to account for site variation.

## 5. Conclusions

In this study, we propose a tree species classification method based on UAV-LiDAR intensity frequency features, i.e., extracting intensity frequency features from LiDAR data and combining the PCS algorithm to segment the acquired point cloud into ITC and the random forest model to classify the main tree species in the study area. In the research process, we extracted the intensity frequency features of different tree species from resampled ITC point clouds of four densities, and then used the RF model to classify the tree species. The research results show that the intensity frequency features of the point clouds of the four densities can achieve higher accuracy in classifying the eight major tree species in the study area, and the overall accuracy of tree species classification can reach more than 84% even if the point cloud density is resampled to 30%. Therefore, the UAV-LiDAR intensity frequency feature can be used for the classification of different tree species. In addition, intensity frequency feature selection is a relevant guarantee to achieve tree species classification accuracy, and mean rank (MR) is a promising method to achieve important intensity frequency feature selection.

In our future work, we will carry out further research in the intelligent segmentation of individual tree crown, and LiDAR intensity frequency combined with hyperspectral remote sensing data to achieve intelligent identification of tree species, which is another hot spot in the current related research.

**Author Contributions:** Project administration, H.D. and X.L.; funding acquisition, H.D. and X.L.; resources, G.Z.; supervision, F.M. and X.L.; investigation, X.L., L.Z., B.Z., J.X. and D.Z.; data curation, B.Z.; conceptualization, Y.G. and H.D.; methodology, Y.G.; software, Y.G.; validation, Y.G.; formal analysis, Y.G.; writing—original draft preparation, Y.G.; writing—review and editing, Y.G. and H.D.; visualization, Y.G. All authors have read and agreed to the published version of the manuscript.

**Funding:** The authors gratefully acknowledge the support of the National Natural Science Foundation of China (U1809208, 32171785, 32201553), Leading Goose Project of Science Technology Department of Zhejiang Province (2023C02035).

**Acknowledgments:** The authors gratefully acknowledge the supports of various foundations. The authors are grateful to the editor and anonymous reviewers whose comments have contributed to improving the quality of this study.

**Conflicts of Interest:** The authors declare that they have no competing interests.

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
