# Peer review of "Tree Species Classifications of Urban Forests Using UAV-LiDAR Intensity Frequency Data"

_remotesensing, doi:10.3390/rs15010110_

Round 1
Reviewer 1 Report (Previous Reviewer 2)
Dear Authors.
When reading your manuscript for the second time I realized that the experimental set-up was not conducted properly concerning both, the intensity frequency measure, and the study of the classification process with LiDAR point clouds of different densities.
Concerning the former, I have serious doubts about the use of absolute instead of relative frequency. The absolute frequency is dependent on the size of the ITC whereas the relative frequency is not (and although you added in the new manuscript the term relative, the way you defined it on line 285 of the manuscript is absolute). So, if the ITCs of the same species have very different sizes the results and conclusions you took may not hold.
Concerning the latter, the problem resides in the fact that the thinning of the point cloud was made after the segmentation process of the high-density point cloud, i.e. after having identified all the individual tree crowns (ITC) using the high-density point cloud. The identification of the ITC is a crucial step for the subsequent process of classification that relies on the intensity frequency within each ITC. To draw the conclusions you have drawn, it would be needed to determine the ITC per point cloud with different densities and not start in the middle of the process. So, the conclusions you drew regarding this subject are not trustable.
Regrettably, I only can support your work for publication if these aspects are revised.
Another important aspect is that you did not address the geometric accuracy issue. In the text, you do not mention that strip adjustment was carried out. Without strip adjustment, the probability to have planimetric shifts (not uniform in all the areas) is high, certainly when you acquired data on different dates. An accuracy assessment, mainly in planimetry should have been conducted. It appears, by one of the answers given to one of my previous questions that the LiDAR data has been shifted after visual inspection. This aspect should also be addressed in the work.
Other comments/suggestions are found in the pdf.

Author Response
Dear reviewer:
I am very grateful to your comments for the manuscript. According with your advice, we amended the relevant part in manuscript. Some of your questions were answered in PDF.

Reviewer 2 Report (New Reviewer)
Abstract: Please modify the sentence or merge the sentences as one sentence is very short "Intensity frequency is defined as the number of 12 times a certain intensity value appears in the ITC point cloud. It is a relative frequency."
Introduction:
1. The purpose of study and contribution of the paper should be explained clearly.
2. Should also elaborate the problem with previously designed methods for classification.
3. Conclusion should be crisp.
4. Few latest references can be added.
5. Modifications in some sentences are required. I would recommend to go through the entire manuscript thoroughly.
Author Response
We appreciate your kind suggestions and we have amended the manuscript accordingly. In addition, the followings are answers for your questions:
1. The purpose of study and contribution of the paper should be explained clearly.
Response: Thank you very much for your comments. We have revised the manuscript.
The revisions are shown below:
“In this study, we propose a tree species classification method based on UAV-LiDAR intensity frequency features”
“Our overall aim was to gain an understanding of the species discrimination potential of intensity frequency features.
The specific research aims were to:
- Describe the differences of intensity frequency between interspecies and intraspecies.
- Demonstrate the ability of intensity frequency feature in tree species classification.
- Quantify the effects of point cloud density in species classification.
- Examine the consistency between feature selected by MR and feature selected by RF.”
2. Should also elaborate the problem with previously designed methods for classification.
Response: Thank you very much for your comments. We have revised the manuscript.
3.Conclusion should be crisp.
Response: Thank you very much for your comments. We have revised the manuscript. The revisions are shown below.
The revisions are shown below.
“In this study, we propose a tree species classification method based on UAV-LiDAR intensity frequency features, i.e., extracting intensity frequency features from LiDAR data and combining PCS algorithm and random forest model to classify the main tree species in the study area. In the research process, we extracted the intensity frequency features of different tree species from the point cloud data of four densities, and then used the RF model to classify the tree species. The research results show that the intensity frequency features of the point clouds of the four densities can achieve higher accuracy in classifying the eight major tree species in the study area, and the overall accuracy of tree species classification can reach more than 85% even if the point cloud density is resampling to 30%.Therefore, the UAV LiDAR intensity frequency feature can be used for the classification of different tree species. In addition, intensity frequency feature selection is an important guarantee to achieve tree species classification accuracy, and mean rank (MR) is a promising method to achieve important intensity frequency feature selection.
In our future work, we will carry out further research in the intelligent segmentation of individual tree crown, and Lidar intensity frequency combined with hyperspectral remote sensing data to achieve intelligent identification of tree species, which is another hot spot in the current related research.”
4.Few latest references can be added.
Response: Thank you very much for your comments. We have renewed some latest references.
5.Modifications in some sentences are required. I would recommend to go through the entire manuscript thoroughly.
Response: Thank you very much for your comments. After through the entire manuscript thoroughly, we have corrected the problematic sentences to the best of our ability.
Reviewer 3 Report (New Reviewer)
Dear authors, I immediately congratulate you on the work done.
I believe that the article is ready to be published, requiring only a small modification in the conclusion topic.
I believe that the conclusion of the article is not adequate to what was presented, I suggest that the authors redo the conclusion topic with texts that present the authors' vision more and not point out results as already stated in the summary topic.
Author Response
Thank you very much for your recognition of our manuscript. We have revised the manuscript according to your suggestions.
1.I believe that the article is ready to be published, requiring only a small modification in the conclusion topic.
Response: Thank you very much for your comments. We have revised the manuscript.
2.I believe that the conclusion of the article is not adequate to what was presented, I suggest that the authors redo the conclusion topic with texts that present the authors' vision more and not point out results as already stated in the summary topic.
Response: Thank you very much for your comments. We have revised the manuscript.
The revisions are shown below.
“In this study, we propose a tree species classification method based on UAV-LiDAR intensity frequency features, i.e., extracting intensity frequency features from LiDAR data and combining PCS algorithm and random forest model to classify the main tree species in the study area. In the research process, we extracted the intensity frequency features of different tree species from the point cloud data of four densities, and then used the RF model to classify the tree species. The research results show that the intensity frequency features of the point clouds of the four densities can achieve higher accuracy in classifying the eight major tree species in the study area, and the overall accuracy of tree species classification can reach more than 85% even if the point cloud density is resampling to 30%.Therefore, the UAV LiDAR intensity frequency feature can be used for the classification of different tree species. In addition, intensity frequency feature selection is an important guarantee to achieve tree species classification accuracy, and mean rank (MR) is a promising method to achieve important intensity frequency feature selection.
In our future work, we will carry out further research in the intelligent segmentation of individual tree crown, and Lidar intensity frequency combined with hyperspectral remote sensing data to achieve intelligent identification of tree species, which is another hot spot in the current related research.”
Round 2
Reviewer 1 Report (Previous Reviewer 2)
See annexe

Author Response
We appreciate your kind suggestions and we have amended the manuscript accordingly. In addition, the pdf are answers for your questions.

This manuscript is a resubmission of an earlier submission. The following is a list of the peer review reports and author responses from that submission.
Round 1
Reviewer 1 Report
This paper describes a method to classify tree species from UAV lidar using the distribution of the intensity of the points. The paper is far from the standard for publication. To be totally transparent I was not even able to read and understand it to the end because the English language and vocabulary are too weak and odd. Because the paper does not even include line numbers, so I will not be able to pinpoint weaknesses precisely but below is a non exhausitive list of problems I enountered:
- The content has not been seriously reviewed. Some English mistakes are something I can understand, but numerous occurrences of additional letters, missing letters, sentences not properly ended with a dot and started with capital letters, random occurrence of text in exponent or uppercase text and so on are the signature of the absence of significant review.
- An example of what I'm mentioning above is the following sentence: "Some cloud versions in this study are ALL LAS1.4". What is a cloud version? Is "ALL" and unknown acronym or authors did not review the text to check if the case was correct? What do authors mean with "some [...] are all [...]"?
- The word "point" and "point-cloud" are used interchangeably to an extent that I was not able to understand the paper. Are we processing points in a point-cloud? Multiple point-clouds? This was very confusing. A point is a sizeless entity with 3 spatial coordinates, a point-cloud is a set of points.
- Figures do not have proper captions. In a scientific paper, figures are expected to be understood standalone. This means they have long captions that describe in depth the content including some legends where necessary. Authors provided figures with very minimal captions that do not meet publication standards. Figure 1, for example, is made of 5 sub-panels and yet the caption has 5 words.
- Bibliography is odd and inconsistent with some citations in authors year style (often duplicated like in "Nicholas et al. (Nicholas et al.,2012)" and sometimes in numeric format. Sometime there are even inconstancies in author year duplications like in "Hamraza et al. (Hamid et al., 2019)". It is like if the text was written by two different people without concerting on the bibliography format and without using bibliography manager. This is a very obvious problem that should have been spotted by the Editors prior to sending this manuscript to review.
- Bibliography elements contain DOI links. Some of them are inccorrect and are pointing to nothing but worse, some of them are pointing to sci-hub, an illegal platform. Using sci-hub is a thing, putting links to sci-hub in you bibliography is another.
- What does "mean of intensity frequency" means? This term is used many times in the paper and is at the core of the study but is meaningless and prevents understanding anything in the paper. I think the authors referred to the mean intensity of the points for a given tree. Not sure.
- Figure 6 contains 64 (64!) diagrams most of them being indistinguishable from each other with respect to the size and resolution of the pictures. They do not bring much information anyway.
- Figure 6, the 32 profiles have all independent and different scales and are thus impossible to compare. This is by far below standard for publication.
- Mathematics is not to the publication standard. Symbols may be inconsitently uppercase or lowercase and with various fonts and various and inconsistent indices. It is hard to follow what we are talking about.
- The word "canopy" is used in an odd manner. Not sure what authors meant and what they are processing. Example among other "The difference in the number of individual canopy point clouds is one of the canopy intensity noises" This sentence is not understandable to me.
- Table 5: I do not think any reader can learn anything from table 5. Moreover, the symbol I, that is used regularly in the paper and is at the core of the study, has never been described properly. I don't know what it refers to and this was absolutely critical for understanding the paper.
- What is a "continuous integer"? Integers are discrete. Real numbers are continuous. A number is not continuous or discrete. It is just a number.
- Sentence: "After assuming that the standard deviation multiple is meanK [...]". What does this mean? What is meanK? Or maybe mean K? The rest of the sentence is of the same kind with missing definitions.
- What are the 147 variables of intensity used. This is not described. One of them is described as the "mean intensity frequency" but see my previous comment about this word.
I'm stopping here even if I could show several more issues. I only chose some of them to illustrate the important issues with the manuscript. The manuscript is far from meeting the standards for publication at a point that I cannot make any comment to the content itself. Yet, I am not saying there no potential. My rough understanding of the paper, based more on guesses than actual understanding, is that authors used, for each segmented tree, the discrete distribution of the intensity of the points to feed a random forest. The discrete distribution is computed using 147 bins that correspond to 147 inputs to the RF. Authors assumed that the RF will be able to discriminate between species using the shape of the histogram. Why not? This may be worth a study and a publication but not in the current state of the paper.
Reviewer 2 Report
- See attachment
